# Persuasiveness of Public Health Communication During the COVID-19 Pandemic: Message Framing, Threat Appraisal, and Source Credibility Effects

**DOI:** 10.3390/ijerph22010030

**Published:** 2024-12-29

**Authors:** Natalia Stanulewicz-Buckley, Edward Cartwright

**Affiliations:** 1School of Psychology, Aston University, Birmingham B4 7ET, UK; 2Leicester Castle Business School, De Montfort University, Leicester LE1 9BH, UK; edward.cartwright@dmu.ac.uk

**Keywords:** health communication, pandemic, loss/gain framing, affect, persuasion, attitude, source credibility, COVID-19

## Abstract

This study examines the relative effectiveness of the UK government’s public health messages used during the first wave of the COVID-19 pandemic. We focus on the use of a loss versus gain frame. We look at the effect of framing on behavioural inclination to follow COVID-19 guidance, as well as affective mechanisms and individual characteristic moderators that might explain said willingness. We ran two studies with a voluntary sample of the UK adult population (total *n* = 300). Across both studies, we only find a significant impact of message framing on the level of negative affect triggered, with the loss frame triggering a higher negative affect. Instead, attitude to public health communication had a direct and indirect effect on behavioural inclination. Our results suggest that threat minimisation and satisfaction with authorities handling a health crisis might be key to consider when developing effective public health communications.

## 1. Introduction

Nothing in the current century has affected so many billions of people globally as the COVID-19 pandemic. This worldwide health crisis has put unprecedented pressure on national health services, as well as resulted in dramatic economic and life losses. Just in the UK, as of 22 December 2022, 177,977 people had lost their lives in relation to being infected with the SARS-CoV-2 [1]. This is despite a national campaign to “limit the spread of the virus and save lives” run by the UK government through the pandemic [2]. National campaigns typically use public health communications (PHCs) as one of the tools that governments or health authorities use to inform the public about the recommended course of action in the face of certain events, such as health or environmental crises. Whether those public health communications are effective in informing people and leading to a high likelihood of them following the recommended course of action cannot be certain though.

It is clear from the previous literature that acting against persuasion, also known as reactance or a boomerang effect, is quite a common reaction to communications that evoke a threat to individual freedom [3]. The UK’s public health communications stressed the importance of “staying at home, to save lives”. Thus, one’s freedom could indeed be seen as targeted and threatened. The effectiveness of such messaging is, therefore, unclear. One of the techniques that has previously been used successfully to promote health-related behaviours, and been demonstrated to have some protection against reactance, is careful framing of messages (e.g., [4,5,6]). We pay particular attention in our work to the loss/gain framing effect. Specifically, the message to ”stay at home” can focus on a positive/gain frame of “saving lives” or a negative/loss frame of “people dying”. Tversky and Kahneman (1981) powerfully demonstrated that perceptions of health interventions were heavily influenced by whether a choice is framed as “people will be saved” or “people will die” [7]. We wished to explore the impact of gain/loss message framing on people’s willingness to act on public health communication guidance in an extraordinary time.

In looking at the impact of gain/loss framing, we recognise that behaving consistently with the public health messaging during the pandemic is a form of public good [8]. Specifically, citizens are asked to sacrifice personal freedoms to stop the spread of the virus and protect others. Andreoni (1995) showed higher levels of cooperation in a public good game when the positive externality of cooperation was emphasized rather than the negative externality of non-cooperation [9]; see also [10,11,12]. There is also extensive evidence that cooperation rates are higher when cooperation is framed in terms of providing a public good rather than maintaining a public good [13,14]. A priori, therefore, it would be reasonable to conjecture that a gain frame that emphasizes the positive externality of contributing to a public good would be most effective at fostering cooperation. The UK government, however, chose to use a loss frame that emphasized the negative externality of not obeying the rules. It is of interest, therefore, to test whether a gain frame would be more appropriate than a loss frame.

With the realisation that due to the interconnectedness of the world regions the future will most likely bring more global or sub-global health crises, it is of great importance to examine (i) the persuasive effects that actual public health communications have on individuals in times of crisis; (ii) what personal characteristics can potentially moderate these effects; (iii) what role affect plays in public health communications persuasiveness; and lastly (iv) whether using alternative public health communications could have resulted in more positive attitudes and higher willingness to follow the governmental guidelines. Thus, the current project aims to further understanding of the persuasive effects of real-life public health communications that were used by the UK government during the pandemic (study 1) and compare them to an alternative framing (study 2). The alternative framing differed from the Government’s in using a positive/gain frame allowing a clear-cut loss/gain distinction [15]. This provided a further test of the loss/gain framing in a novel context, as well as to aid understanding of potential moderating (i.e., threat appraisal, and source credibility) and mediating factors (i.e., positive and negative affect (PA and NA)).

We proceed as follows. In Section 2, we discuss the relevant literature and the underlying psychological model we will consider. In Section 3, we describe our quasi-experimental methods. In Section 4, we provide the results from study 1 and in Section 5 we provide the results from study 2. In Section 6, we conclude.

## 2. Framing Effect and the COVID-19 Pandemic

Research on the evaluation of public health campaigns has been developing slowly [16]. Public health communications that apply loss and gain framing have been examined widely, among others, in topics such as antidrug campaigns [17,18], promoting healthy eating behaviours [19], exercise [20], smoking cessation [21], and vaccinations [22]. But their role during a global health crisis has just started to be examined. Framing effects can be understood as the impact that slight variations in the language used in a communication have on the perception and compliance with said message. Typically, following from the work of Rothman and Salovey (1997), message framing is divided into loss focused versus gain focused [15]. The loss framing highlights the negative outcomes that can occur should an individual decide not to comply with the guidelines presented in a message, whereas gain framing focuses on stressing the benefits that might happen should one comply (e.g., [23]). For example, “if you stay at home, you will protect the NHS (National Health Service), and people will live” would be an example of a gain-focused message, whereas its counterpart “if you do not stay at home, you will not protect the NHS, and people will die” would be using a loss frame. Which framing would present higher effectiveness is not clear however and requires testing. Indeed, the empirical evidence is mixed when it comes to assessing which type of framing is more effective in triggering audience’s compliance with health communications. Some studies show gain-frame messages as more effective (e.g., [24] (for prevention behaviour); [25,26,27]), while others demonstrate the opposite (e.g., [28,29]).

A few papers where loss and gain framing have been utilised and tested in the context of health-protective behaviours related to the current COVID-19 pandemic have been published (e.g., [30,31,32,33]). In one such paper, the authors focused on examining which type of message was perceived as more persuasive for oneself versus others ([32]; Colombian sample). It was found that the gain frame was perceived as stronger and more motivating to perform self-care behaviours. This study, however, utilised a selection of self-made adverts, consisting only of white text on a black background, and thus somewhat lacked real-life context. In a similar vein, Doerfler et al. (2021; US sample) [31] used hypothetical COVID-19 scenarios to test gain and loss framing. Here, it was observed that a loss scenario (i.e., lives lost) led to people being more risk seeking. In another study, with a Chinese sample [30], participants were presented with news articles created for the study (partially based on real news) that showed information on COVID-19 vaccination. The framing did not demonstrate a significant effect on vaccination intentions or attitudes. Finally, researchers [33] (US sample) applied hypothetical scenarios, focused on shop signs promoting mask wearing, and found somewhat mixed results, but it was clear that in high-risk public environments (specifically, attending a worship with 500+ worshippers), gain-framed messages were always more persuasive. From the above, it should be clear that there is a lack of published studies that have examined the effects of real-life COVID-19 public health communications and that the results reported in relation to framing are mixed. The lack of studies on the effectiveness of real-life public health communications is particularly worrying, because such communications have more complex structures than framing manipulations that are often used in experiments (e.g., they often consist of text and visual stimuli) [34]. This gap in the literature highlights the need for research which uses actual public health communications and underlies the motivation of our study 1.

### 2.1. Role of Affect

It needs highlighting that even though loss-framed messages sometimes seem more persuasive (e.g., [28,29]), they also have been perceived as inherently more threatening [35] and shown to induce more fear [36]. This is why many researchers have suggested that loss framing leads to more reactance (e.g., [5,35,37,38,39]), and thus decreases rather than increases the adherence to guidance presented in such way. As fear is one of the negative emotions, negative affect might be particularly important to consider as the driving force behind the loss-frame effects. Previous research has showed that positive affect (PA) mediates the relationship between a gain frame and message acceptance, whereas negative affect (NA) is more likely triggered by a loss frame, but it can lead to a higher intention to engage in healthy behaviour [40]. The role of affect in the persuasive effects of public health communications was also demonstrated by a recent meta-analysis [41]. The summarized evidence showed that a gain frame triggers PA, while a loss frame induces NA, both of which in turn affect the effects of the frames.

Due to the above, we examined the role of both PA and NA as mechanisms behind higher versus lower willingness to follow COVID-19 guidance. This is a novel context in which to be testing the role of affect on persuasion. We expected NA to lead to decreased rather than increased willingness to follow public health communications, and the opposite for PA.

### 2.2. Potential Moderators: Threat Appraisal and Source Credibility

When considering when and for whom public health communications work (also known as their moderators), previous studies have predominantly focused on the factors that relate to risk perception [42]. This is guided by Prospect theory and the notion that loss-framed messages encourage engagement in behaviours that are construed as risky, whereas gain-framed messages motivate engagement in behaviours that are construed as safe [43,44]. The second area that has gathered a lot of interest relates to the source of the message itself as a factor affecting its persuasiveness (e.g., [45,46]). There are of course other factors, e.g., people’s sensitivity to favourable/unfavourable outcomes (such as BIS/BAS) but they were not considered in the current study as the previous evidence suggests they are not highly impactful (apart from for young people, who might have less developed health knowledge) [42].

Thus, when selecting potential moderators to consider we were guided by risk and the message source. From the threat appraisal perspective, we included perceived susceptibility to COVID-19 and threat minimisation as the variables of interest (we did not include other related variables, such as risk severity, the second element of risk in Protection Motivation Theory [47,48] due to the time constraints, as well as the assumption that with the widespread information campaign, that aspect would be similar for our participants). COVID-19 susceptibility represents risk/threat perception, while threat minimisation can be seen as a coping mechanism (i.e., minimising the threat that was not much understood at the beginning of the pandemic). Threat appraisal and coping appraisal are the two key elements of Protection Motivation Theory [47,48]; they are seen as drivers of health-related behaviours. The second broad aspect we included regards factors linked to the source of a message, here, the UK government, which was the source of the public health communications chosen for the study. The long-standing line of research demonstrated that source credibility (stemming from expertise, trustworthiness, and/or likability) affects audience’s response (Source Credibility Theory [49]), with more credible sources being more persuasive [50,51]. Thus, following from this, we included satisfaction with government handling of the COVID-19 pandemic (study 1), as well as trust in the government, past compliance with governmental policies, and lastly political orientation (study 2) as proxy measures of source credibility.

### 2.3. The Current Study

The aims of the current study were (1) to examine the emotional and persuasive effects of exemplar public health communications related to the COVID-19 pandemic that were released by the UK government; and (2) whether manipulating the framing of those messages improves said effects. We also (3) aimed at examining individual differences that might significantly affect one’s reactions to public health communication. This is why, in study 1, we examined reactions to two posters published by the UK government urging people to stay at home during the COVID-19 pandemic (which assured high ecological validity), whereas in study 2 we adapted one of those messages, to be able to compare the effects of a more clear-cut gain versus loss-focused frame, and explored the role of individual characteristics for their effectiveness. The following hypotheses were developed for this project.

**H1.** 
*More positive (study 1) or gain-framed (study 2) public health communication: The more positive the attitude, the more/less the positive/negative affect, and the higher the inclination to follow the guidance presented in the HPC.*


**H2.** 
*Both PA (positively) and NA (negatively) will mediate the effect of the public health communication attitude on one’s behavioural inclination to follow COVID-19 guidelines.*


Furthermore, in line with previous research suggesting the role of certain individual characteristics affecting people’s perceptions and reactions to persuasive messages in general, we also examined the role of certain individual characteristics (described above) on the persuasiveness of the public health communications.

**H3.** 
*More/less negative public health communications’ effects on public health communication reactions (NA, PA, and public health communication attitude) will be moderated by individual characteristics, with more negative public health communication showing a more negative effect and strengthening with high COVID-19 susceptibility, low government satisfaction, and low threat minimisation.*


**H4.** 
*The loss vs. gain frame effect on public health communication reactions (NA, PA, and public health communication attitude) will be moderated by individual characteristics, with a loss frame showing a more negative effect, and strengthening with low threat minimisation, high trait reactance, high COVID-19 susceptibility, and low level of governmental support (i.e., low satisfaction with government’s handling of the pandemic, low past compliance with government policies, and political orientation opposite to the current government).*


## 3. Materials and Methods

### 3.1. Study 1

Participants: A total of 138 UK participants (mean age = 32.8 ± 11.64 SD; 81% female; 59% with higher education degree; 94% with English as native language) were recruited from the Prolific participant pool and social media. To take part in the study people were required to be UK residents. This was a voluntary sample of UK residents (ibidem for study 2), recruited from Prolific. Written informed consent was obtained from each participant. The study was approved by the De Montfort University Research Ethics Committee. The study was run online using the Qualtrics platform. All data collection occurred in May–July 2020 (the first pandemic wave). The participants that took part via Prolific were compensated for their time (£1.75). The survey included an attention check. All participants who failed it were removed from the analysis. The sample size was based on the consideration of analyses planned, but also practical considerations (available funds and time urgency for data collection in the early pandemic). As the main interest of this study was an analysis of differences between more negative/loss and more positive/gain messages, we ran a power analysis to estimate the recommended sample size. The analysis with the power = 0.80, alpha = 0.05, and small-to-medium effect size (Cohen’s d = 0.40), suggested a sample of 156 participants, which we aimed to recruit for both studies.

Design: This study used a between-subjects quasi-experimental design, where participants were randomly introduced to one of two public health communications; see Figure 1A. Both of these messages were taken from the governmental social media accounts, and represented communications aimed at promoting staying at home during the COVID-19 pandemic. They had the same background but differed in the text used, with one being more negative (“If you go out, you can spread it. People will die”, *n* = 66) in tone than the other (“Anyone can get it. Anyone can spread it”, *n* = 72). They both also had a solution presented stating “Stay home to help us save lives”.

Measures: Questionnaires were used to measure variables of interest, first we describe those that assessed reactions to public health communications; second, the ones that were regarding individual differences of interest.

Message attitude: The attitude towards public health communication was measured using a scale developed by Laczniak and Teas [52]. This measure consists of 14 pairs of contrasting adjectives that can be used to describe someone’s attitude toward an object, for example, irritating–not irritating. Participants used a Likert-style scale from 1 (closest to the left-hand-side option) to 5 (closest to the right-hand-side option) to indicate their opinion regarding the public health communication that they were presented with. The scale showed good reliability, with Cronbach’s alpha = 0.88.

Emotions: The emotions experienced in response to seeing the public health communication were measured using the Positive and Negative Affect Schedule (PANAS-SF; [53]). Here, participants were asked to indicate how strongly they felt by responding to each of 20 emotional adjectives, using a 1 (“very slightly or not at all”) to 5 (“extremely”) Likert-style scale. Ten adjectives were then used to create a score for positive affect (PA), whereas the other 10 provided the score for negative affect (NA). This measure has demonstrated good reliability and validity previously (e.g., [54,55]). In the current study, reliability was also very good (Cronbach’s alpha = 0.89 for both).

Behavioural inclination: After seeing the public health communication, participants were also asked two questions regarding the behavioural effects of the said message: (1) How likely are you to follow the advice to stay at home after seeing this poster?; and (2) How would you say did this poster affect your attitude to follow the guideline to stay at home? The first question had the following answer options: not at all likely (1), a little likely (2), somewhat likely (3), mostly likely (4), and extremely likely (5). Whereas the second question could be answered with the following: decreased it a lot (1), decreased it a bit (2), neither decreased, nor increased (3), increased it a bit (4), and increased it a lot (5). The two items were averaged to create a behavioural inclination score for each individual; they were highly and positively correlated (r = 0.48, *p* < 0.001), demonstrating good reliability of this index.

We turn now to measures of individual differences. Perceived susceptibility to COVID-19: The level to which participants believed they were vulnerable to catching COVID-19 was measured with four items: “I feel really susceptible to becoming infected with coronavirus”, “I don’t think there is a substantial chance of me catching coronavirus”, “I feel like I am extremely well protected from becoming infected with coronavirus”, “I worry that I will definitely catch coronavirus”. These were answered on a 1 (disagree strongly) to 5 (agree strongly) Likert-style scale. The reliability analysis showed satisfactory reliability level (Cronbach’s alpha = 0.79).

Threat minimization: We measured the level to which participants minimised the threat of the current pandemic. This was measured with four items: “The seriousness of the coronavirus situation in the UK has been overblown by most people”, “The coronavirus crisis in the UK is worrying me a lot”, “The coronavirus situation in the UK doesn’t differ from any other health issue (e.g., seasonal flu)”, “I find myself panicking a lot about the coronavirus situation in the UK”. Participants used a 1 (disagree strongly) to 5 (agree strongly) Likert-style scale to respond. This measure demonstrated adequate reliability, although it could be improved (Cronbach’s alpha = 0.59).

Satisfaction with government handling of COVID-19 pandemic: This construct was measured with one item: “In general, how much are you satisfied with the way UK government has been dealing with the COVID-19 pandemic?”. It was measured on a 1 (not at all) to 5 (extremely) Likert-style scale, with a higher score representing a higher level of satisfaction. A similar way of measuring this variable was reported previously (e.g., [56]).

Demographics: Participants were asked to record their age, sex, education level, native language, whether they thought they contracted the virus, and whether they thought a member of their family had experienced COVID-19.

Procedure: The study took place online, on the Qualtrics platform. All participants were presented with an information sheet to start with. If they decided to take part, they progressed to the questionnaire. First, they saw one of the public health communications and were asked questions about it (questions about PA, NA, message attitude, behavioural inclination), after which they were taken to questions measuring individual characteristics (display order of those was randomised). Lastly, they provided demographics and indicated their up-to-date COVID-19 protective behaviours as well as the presence of symptoms in them or their family. When the public health communication was displayed, options to move to another screen were disabled, to increase the chances of engagement with the stimulus. Participants had to complete the study in one setting. At the end they were debriefed and provided with the contact details of the researchers. Participants recruited via Prolific were reimbursed for their time.

Data collation: The data were exported to the SPSS version 25 software, where all the data checks were performed, together with reverse-scoring specified items and computing the variables of interest.

### 3.2. Study 2

Participants: A total of 158 UK participants (mean age = 32.47 ± 10.89 SD; 70% female; 59% with higher education degree; 86% with English as native language) were recruited from the Prolific participant pool (2 extra participants were dropped as they were not UK residents). To take part in the study they were required to be UK residents. Written informed consent was obtained from each participant, who were compensated for their time. The study was approved by the De Montfort University Research Ethics Committee. The study was run online using the Qualtrics platform. All data collection occurred in July 2020. The survey included an attention check. All participants who failed it were removed from the analysis.

Design: This study used a quasi-experimental between-subjects design, where participants (*n* = 80 in loss group, *n* = 78 in gain group) were randomly introduced to one of two public health communications. The first of the messages was the same one as used in study 1 (i.e., the one with loss-framed/more negative text), whereas the second one was adapted by the authors to represent a gain frame; see Figure 1B. As you can see, the text has been changed to provide a positive/gain frame, emphasizing “stay at home… people will live”.

Measures: Message attitude: The attitude towards public health communications was measured with the same questionnaire as in study 1. The reliability was very good (Cronbach’s alpha = 0.87). 

Emotions: The emotions experienced in response to seeing the public health communication were measured with the same questionnaire as in study 1. The reliability was very good (PA Cronbach’s alpha = 0.87; NA Cronbach’s alpha = 0.89).

Behavioural inclination: After seeing the assigned public health communication, participants were also asked about their inclination to engage in behaviours protecting them from COVID-19 with the following five items: “After seeing this advert I intend to: (1) Stay at home as much as I can; (2) Avoid going out for non-essential purposes (e.g., other than essential food, medicines); (3) Keep physical distance if I have to go out; (4) Avoid social gatherings with friends; (5) Avoid crowded places”. Participants indicated their responses using a 1 (strongly disagree) to 5 (strongly agree) Likert-style scale. The reliability of this scale was excellent, with Cronbach’s alpha = 0.90.

Individual differences. Political orientation: We used one previously published question [57] to assess participants’ political orientation. Here, participants had to choose one label that reflected their political standing in the most accurate way. The options were as follows: (1) very liberal; (2) liberal; (3) middle of the road; (4) conservative; and (5) very conservative. This question has been shown to be a reliable measure of political attitudes previously [58].

Trust in the government: This construct was measured with a four-item scale [59]: (1) Do you think that you can trust the government to do what is right? (2) Do you think that people in government waste a lot of the money we pay in taxes? (3) Would you say that the government is pretty much run by a few big interests looking out for themselves, as opposed to being run for the benefit of all the people? (4) Do you think that quite a few of the people running the government are crooked?. It was measured on a 1 (not at all) to 5 (very much) Likert-style scale, with a higher score representing a higher level of trust. The reliability and validity of this scale has been demonstrated in the literature [59]. In the current study, its reliability was satisfactory (Cronbach’s alpha = 0.83).

Citizen compliance: Lastly, we measured general compliance with governmental policies. This was achieved with the following items: (1) I am willing to comply with government policies even if those policies conflict with my own interest; (2) I tend to follow what government wants me to do; (3) I often find myself disagreeing with the policies introduced by the government. This scale was adapted from Im and colleagues’ work [59]. Participants used a 1 (not at all) to 5 (very much) Likert-style scale to present their opinions. In the current study, the reliability of this scale was satisfactory (Cronbach’s alpha = 0.66).

Perceived susceptibility to COVID-19. This was measured in the same way as in study 1 and demonstrated good reliability (Cronbach’s alpha = 0.71). Threat minimisation was also measured in the same way as in study 1 and demonstrated good reliability (Cronbach’s alpha = 0.70).

Procedure: The same procedure as in study 1 was applied. 

Data collation: The data were exported to the SPSS version 25 software, where all the data checks were performed, together with reverse-scoring specified items and computing the variables of interest.

## 4. Results of Study 1

For both studies, details regarding participant characteristics and descriptive statistics are reported, together with the reliability index of each measure. This is followed by the comparison of the reactions to the public health communications used in the studies, as well as an examination of the relationships between study variables. Lastly, mediation and moderation analyses are reported. All the analyses have been selected in line with the purpose of this study, specifically, comparing responses to public health communications, and examining the moderating effect of individual characteristics, as well as the mediating effect of affect.

### 4.1. Descriptive Statistics—Study 1

Table 1 summarizes the overall statistics regarding safety behaviours undertaken by the participants, as well as other COVID-19 outcomes, such as experiencing symptoms of COVID-19, satisfaction with the government’s handling of the COVID-19 pandemic, and behavioural aspects linked to the public health communication observed. It can be seen that overwhelmingly (>90%) participants reported following the safety guidance regarding hand washing, social distancing, and going out only for essential reasons. Among the sampled population, just under 20% reported experiencing COVID-19 symptoms, with a slightly higher number (~25%) for seeing those symptoms in the family. Only around 30% of the participants were not satisfied with the government’s handling of the current pandemic, with others showing at least some level of satisfaction.

Lastly, regarding the behavioural inclination after seeing the study’s public health communications, overwhelmingly participants reported a somewhat high likelihood of following the guidance (~82%) recommended in there, with a slightly smaller number suggesting that their inclination to follow the advice somewhat increased (~56%) after seeing the public health communication. Table 2 presents the mean, SD, minimum, and maximum scores of the variables of interest obtained in study 1, divided into those assessed in relation to seeing public health communications, as well as some individual characteristics. The reliability index is also reported. It is worth highlighting that all measures apart from threat minimisation tendency showed satisfactory reliability (although a Cronbach’s alpha equal to 0.59 can be considered acceptable given this is the first time this measure was used).

### 4.2. Message Reactions by Condition

To examine, whether there were differences in the way participants reacted to the two public health communications used in study 1, a series of independent group *t*-tests was run. The results (Table 3) showed no significant differences in any of the measured outcomes, that is, in message attitude (*p* = 0.72), positive (*p* = 0.77) or negative (*p* = 0.40) affect experienced, or behavioural inclination to follow the guidance regarding staying at home (*p* = 0.33). Due to this, in what follows, both public health communication conditions were collapsed and treated as a single group.

### 4.3. Correlations Between Study 1 Variables

A series of partial correlations (controlling for the public health communication condition) was run to establish the relationships between the message reaction outcomes (message attitude, PA, NA, and behavioural inclination) and individual characteristics of participants. The results are displayed in Table 4 and show that the most consistent pattern of relationships was observed for satisfaction with the government’s handling of the COVID-19 pandemic (which was significantly related to all the outcomes), followed by threat minimisation (significant relationships apart from for PA). Overall, these results show that higher satisfaction with the government’s handling of the current pandemic, as well as lower threat minimisation of the pandemic, affect one’s reactions to public health communication, including most importantly one’s behavioural inclination.

### 4.4. Parallel Multiple Mediation Model

Due to the public health communications used in study 1 being assessed as similar (see Table 3 showing no significant differences in their perception) by participants, rather than testing mediation mechanisms with the public health communication condition (i.e., more vs. less positive) as the predictor variable, the message attitude was used as the predictor variable instead. Message attitude has been suggested as an important factor predicting people’s intentions and behaviours previously (see Theory of Planned Behaviour [60]). Thus, here, we analysed whether the effect of message (public health communication) attitude on the behavioural inclination to follow COVID-19 guidance was mediated via affect (PA and NA). The parallel mediation model was run in the Process macro (model 4, with 5000 bootstrapping samples) [61]. Because both PA and NA were expected to show an effect, they were entered simultaneously as mediators. The unstandardised model coefficients are presented in Figure 2a. Regressing NA on message attitude showed that a more positive message attitude significantly decreased NA (a_1_ = −0.46 (0.11); *p* < 0.001; 95% CI: −0.67, −0.24). Regressing PA on message attitude showed that a more positive message attitude increased PA (a_2_ = 0.68 (0.10); *p* < 0.001; 95% CI: 0.48, 0.88). Finally, regressing behavioural inclination on NA, PA, and message attitude, showed that both mediators significantly affected the outcome, and that a significant direct effect of message attitude on behavioural inclination remained significant after the mediators were modelled (b_1_ = 0.27 (0.07); *p* < 0.001; 95% CI: 0.12, 0.41; b_2_ = 0.22 (0.08); *p* = 0.005; 95% CI: 0.07, 0.37). A mediation analysis showed that both NA (a_1_b_1_) and PA (a_2_b_2_) significantly mediated the effects of the message attitude on behavioural inclination to follow COVID-19 guidance (a_1_b_1_ = −0.12 (0.04); 95% CI: −0.21, −0.04; a_2_b_2_ = 0.15 (0.05); 95% CI: 0.05, 0.26). This suggests that a more positive message attitude was related to a higher level of PA and lower level of NA experienced, which in return affected one’s inclination to follow health guidance. Lower NA and higher PA were related to more positive behavioural intention to follow COVID-19 guidance. This indicates that messages that trigger positive attitude, low NA but high PA might be the most effective in promoting positive behavioural inclination to follow COVID-19 guidance. Overall, R^2^ = 41.49% of the variance in behavioural inclination to follow COVID-19 guidance was explained by variables in the model. Proportion-mediated (PM) effect size estimates for the individual mediation pathways indicated that the NA pathway (PM = 20%) conveyed a slightly smaller portion of the impact of message attitude on behavioural inclination, compared with the PA pathway (PM = 24%). This suggests that there are other mediators (than affect) that could contribute to the understanding of effects of public health communications on people’s behavioural inclination to follow health guidance.

The following equations describe the parallel multiple mediation model below (see Figure 2a).

**Figure 2 ijerph-22-00030-f002:**
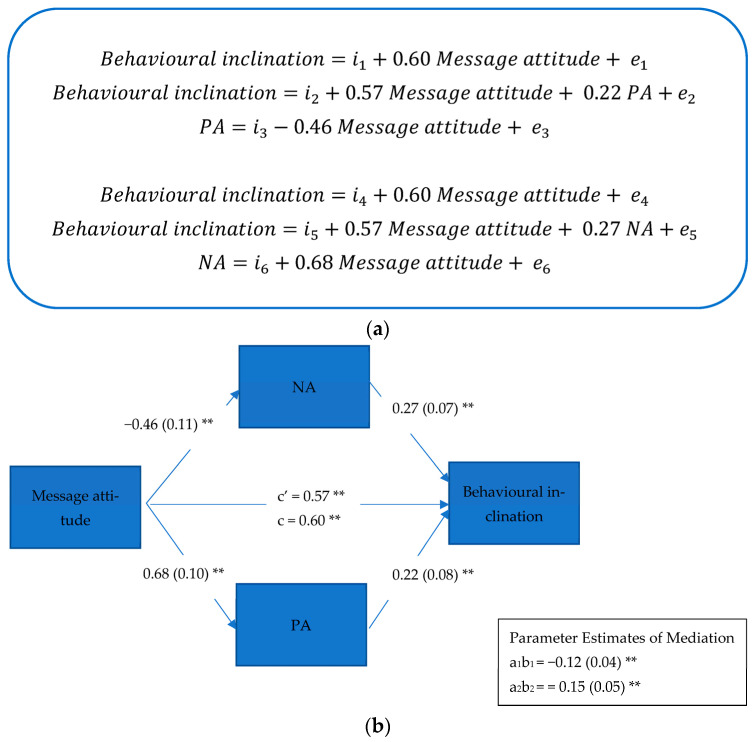
(**a**) Mediation equations from study 1; (**b**) parallel multiple mediator model, estimating affective mechanisms between message attitude and behavioural inclination to follow COVID-19 guidance. ** *p* < 0.001.

### 4.5. Moderation Effects

Lastly, we tested whether the moderating variables examined in this study would affect the relationship between message attitude and public health communication appraisal (i.e., NA, PA, and behavioural inclination to follow COVID-19 guidance). We did not test the public health communication condition as the predictor in the moderation analyses due to the absence of any significant differences between the two groups. We expected that the negative relationship between the message attitude and both NA and willingness to follow COVID-19 guidance would be strengthened with increasing levels of (i) COVID-19 susceptibility, and (ii) dissatisfaction with the government’s handling of the pandemic, as well as decreasing levels of (iii) threat minimisation (and opposite for PA). The moderation analyses were performed using the Process macro [61], with age and gender used as covariates.

In terms of message attitude x COVID-19 susceptibility: (1) Outcome—behavioural inclination to follow COVID-19 guidance. When examining the effect of COVID-19 susceptibility on the link between message attitude and behavioural inclination to follow COVID-19 guidance, we did not observe a significant moderation (b = −0.15, SE = 0.10, t = −1.46, *p* = 0.15, 95% CI = −0.36; 0.05). There was not a direct effect of COVID-susceptibility either (b = 0.08, SE = 0.07, t = 1.15, *p* = 0.25, 95% CI = −0.06; 0.21), but message attitude (b = 0.60, SE = 0.09, t = 6.43, *p* < 0.001, 95% CI = 0.41; 0.78) showed a significant relationship with willingness to follow COVID-19 guidance. (2) Outcome—NA. The same analysis as above with NA as outcome showed no moderation (b = 0.04, SE = 0.14, t = 0.26, *p* = 0.80, 95% CI = −0.24; 0.31), but a significant direct effect of both public health communication attitude (b = −0.46, SE = 0.11, t = −4.20, *p* < 0.001, 95% CI = −0.68; −0.24) and COVID-19 susceptibility (b = 0.19, SE = 0.08, t = 2.33, *p* = 0.02, 95% CI = −0.03; 0.34). (3) Outcome—PA. The same analysis as above with PA as outcome showed no moderation (b = 0.11, SE = 0.12, t = 0.95, *p* = 0.34, 95% CI = −0.12; 0.34), nor a COVID-19 susceptibility effect (b = −0.01, SE = 0.07, t = −0.18, *p* = 0.86, 95% CI = −0.16; 0.13), but a significant direct effect of public health communication attitude (b = 0.68, SE = 0.10, t = 6.67, *p* < 0.001, 95% CI = 0.48; 0.88).

In terms of message attitude x threat minimisation: (1) Outcome—behavioural inclination to follow COVID-19 guidance. A similar analysis with threat minimisation as the moderating variable being tested (see Figure 3) demonstrated a significant interaction effect (b = −0.24, SE = 0.08, t = 3.24, *p* = 0.004, 95% CI = −0.40; −0.08), as well as a significant direct effect (b = 0.26, SE = 0.08, t = 3.24, *p* < 0.002, 95% CI = 0.10; 0.42). Message attitude also showed a significant direct effect (b = 0.48, SE = 0.09, t = 5.62, *p* < 0.001, 95% CI = 0.31; 0.65) on behavioural inclination to follow guidance. The conditional effects are displayed in the top half of Table 5 and show that the strongest negative relationship between message attitude and behavioural inclination to follow guidance was among those with a low level of threat minimisation.

(2) Outcome—NA. The same analysis as above with NA as outcome showed no moderation (b = 0.03, SE = 0.11, t = 0.26, *p* = 0.79, 95% CI = −0.18; 0.24), but a significant direct effect of both public health communication attitude (b = −0.52, SE = 0.11, t = −4.71, *p* < 0.001, 95% CI = −0.73; −0.30) and threat minimisation (b = 0.36, SE = 0.10, t = 3.57, *p* < 0.001, 95% CI = 0.16; 0.56). (3) Outcome—PA. The same analysis as above with PA as outcome showed no moderation (b = 0.11, SE = 0.10, t = 1.06, *p* = 0.29, 95% CI = −0.09; 0.30), nor threat minimisation effect (b = 0.03, SE = 0.10, t = 0.33, *p* = 0.74, 95% CI = −0.16; 0.23), but a significant direct effect of public health communication attitude (b = 0.69, SE = 0.11, t = 6.57, *p* < 0.001, 95% CI = 0.48; 0.90). The results of the moderation analyses for significant interaction effects are presented below, in Table 5.

In terms of message attitude × satisfaction with government handling of COVID-19 pandemic. (1) Outcome—behavioural inclination to follow COVID-19 guidance. A similar analysis with satisfaction with government as the moderating variable being tested (see Figure 4) demonstrated a significant interaction effect (b = −0.31, SE = 0.07, t = −4.12, *p* < 0.001, 95% CI = −0.46; −0.16), even though it did not show a significant direct effect (b = 0.06, SE = 0.05, t = 1.17, *p* = 0.24, 95% CI = −0.04; 0.17). Message attitude showed a significant direct effect (b = 0.62, SE = 0.09, t = 6.89, *p* < 0.001, 95% CI = 0.44; 0.79) on behavioural inclination to follow guidance. The conditional effects are displayed in the bottom half of Table 5 and show that the strongest negative relationship between message attitude and behavioural inclination to follow guidance was among those with a low level of government satisfaction.

(2) Outcome—NA. The same analysis as above with NA as outcome showed a significant moderation (b = −0.24, SE = 0.10, t = −2.53, *p* = 0.01, 95% CI = −0.43; −0.05), as well as a significant direct effect of public health communication attitude (b = −0.40, SE = 0.11, t = −3.51, *p* < 0.001, 95% CI = −0.62; −0.17) but no direct effect of satisfaction with the government’s handling of the COVID-19 pandemic (b = −0.08, SE = 0.07, t = −1.16, *p* = 0.25, 95% CI = −0.20; 0.05). This interaction is presented in Figure 5 and Table 5 and showed that a more positive public health communication attitude was linked to less NA, especially among those with high government satisfaction.

(3) Outcome—PA. The same analysis as above with PA as outcome showed no significant moderation (b = −0.13, SE = 0.09, t = −1.57, *p* = 0.12, 95% CI = −0.30; 0.03), but a significant direct effect of both public health communication attitude (b = 0.58, SE = 0.10, t = 5.70, *p* < 0.001, 95% CI = 0.38; 0.78) and (at trend level) satisfaction with the government’s handling of the COVID-19 pandemic (b = 0.12, SE = 0.06, t = 1.96, *p* = 0.053, 95% CI = −0.001; 0.24).

## 5. Results for Study 2

Following on from and expanding on study 1, all the analyses were selected in line with the purpose of this study, specifically, comparing responses to public health communications, and examining the moderating effect of individual characteristics, as well as the mediating effect of public health communication attitude and affect. Table 6 depicts some basic characteristics of the participants. First, in terms of political orientation, the sample was mostly liberal (~48%) or middle of the road (~39%) in their views, with the great majority not supporting the Conservative party in the last election (72%). Once again, the overwhelming majority reported following the safety guidance around COVID-19, with high levels of compliance for hand washing (~87%), social distancing (~92%), and going out only for essential reasons (~81%). Among the sampled population, similarly to study 1, around 16% reported experiencing COVID-19 symptoms, with a slightly higher number (~23%) for having those symptoms in the family. Only around 26% of the participants were not satisfied with the government’s handling of the current pandemic, with others showing at least some level of satisfaction, again similarly to what was observed in study 1.

### 5.1. Descriptive Statistics—Study 2

The frequencies for the variables of interest in study 2 are presented above, in Table 6.

Table 7 presents the mean, SD, minimum, and maximum scores on the variables of interest obtained in study 2, divided into those assessed in relation to seeing public health communications, as well as some individual characteristics. The reliability index is also reported. It is worth highlighting that all measures showed satisfactory reliability.

### 5.2. Message Reactions by Condition

In order to examine the reactions of participants to our two public health communications, that is, the loss- and gain-framed ones, a series of independent t-tests as well as a non-parametric test (for behavioural inclination due to skewed data) were performed. The results showed that the only significant difference was in the level of NA experienced after seeing the messages (see Table 8 for more details), with those in a loss frame reporting a significantly higher NA level, than those in a gain frame (*p* < 0.001).

### 5.3. Recoding of the Behavioural Inclination to Follow COVID-19 Guidance

It is worth highlighting that even though behavioural inclination to follow COVID-19 guidance in study 1 was analysed with linear regression. In study 2, due to the non-normal distribution and highly skewed data of this main outcome (i.e., behavioural inclination to follow COVID-19 guidelines, which demonstrated a negative binomial distribution with overdispersion of the highest possible value), for the subsequent analyses this variable was recoded into a binary variable, with those who scored the maximum possible value (5) coded as one group (dummy-coded as 1), whereas those who scored above the midpoint (from 3.20 to 4.80) but below the maximum as the second group (dummy-coded as 0). As the public health communications used in the study were motivated by the intention to increase people’s willingness to follow COVID-19 guidance, comparing those two groups, who indicated somewhat (group “0”) or strong (group “1”) intentions to follow said guidance after seeing the public health communications is in line with this study’s primary intent. It is worth stating that this recoding resulted in the exclusion of eight participants, which equates to only 5.1% of all data (seven out of these data points were indicated as outliers by a boxplot analysis).

### 5.4. Correlations Between Study 2 Variables

A series of correlations was run to examine the relationships between the study variables, with special emphasis on what individual factors predict people’s PA, NA, message attitude, and behavioural inclination to follow COVID-19 guidelines (this variable was dummy-coded as “0” and “1” here) after seeing the public health communication. Partial correlations were used, controlling for the public health communication condition. The results showed that PA correlated positively with government compliance, and negatively with threat minimisation, as well as showing positive relationships at trend level with government trust and susceptibility (see Table 9 for full details). NA showed positive correlations with COVID-19 susceptibility and political orientation, as well as a negative relationship with threat minimisation. Public health communication attitude was positively correlated with government compliance and PA, as well as negatively related to trait reactance and NA. Lastly, behavioural inclination to follow COVID-19 guidelines showed positive correlations with governmental compliance, PA, and public health communication attitude.

### 5.5. Mediation Model

Following on from study 1, we examined whether loss/gain framing, as well as public health communication attitude, had a direct effect on willingness to follow COVID-19 guidance, and whether PA and/or NA showed indirect effects. This was investigated in a binary logistic regression (age and gender were kept as covariates), where loss/gain framing was entered in the firsts step, followed by entering public health communication attitude in the second step, and then followed by adding NA and PA in the subsequent step. The results showed that loss/gain framing had no significant effect, whereas public health communication attitude did (see Table 10). The significant effect of public health communication attitude disappeared however after entering affect variables, suggesting that the direct link between public health communication attitude and willingness to follow COVID-19 guidance was fully mediated by PA. NA did not show a significant effect. This replicates to a degree the mediation model tested in study 1.

### 5.6. Moderation Effects

Even though study 2 deliberately adapted one of the public health communications to make a clear difference between loss- vs. gain-framed public health communications, participants did not report much difference in appraisals between the two (only in NA). This is why, for moderation analyses we focused on the public health communication condition as the predictor, NA as the outcome, and individual characteristics as the potential moderators (with age and gender as covariates). The moderators were set using the 16th, 50th, and 84th percentiles. The results of those analyses are presented in Table 11.

As presented in Table 11, the only significant moderating effect was displayed by threat minimisation, where the link between public health communication condition and NA was the strongest for those with a low level of threat minimisation (coeff. = −0.75, SE = 0.17, t = −4.40, *p* < 0.001, 95% CI = −1.09; −0.42), followed by those with a medium level (coeff. = −0.50, SE = 0.12, t = −4.18, *p* < 0.001, 95% CI = −0.74; −0.27) (see Figure 6), whereas for people with a high level of threat minimisation, the link between public health communication condition and NA was non-significant (coeff. = −0.20, SE = 0.17, t = −1.15, *p* = 0.25, 95% CI = −0.53; 0.14). This shows that framing did not have an effect on the level of NA triggered in individuals that scored high on threat minimisation.

## 6. Discussion

Even though public health communications are used widely, research on the evaluation of public health campaigns has been developing slowly [16]. This is a worrying observation, mostly because public health communications are commonplace, and have been highly utilised during the current global health crisis. The COVID-19 pandemic has highlighted the need for more research that looks into the topic of public health communication evaluation, as only then can governments or other advisory bodies produce evidence-based and effective public health communications. Following on from that, the current project examined the perception and effectiveness of public health communications used during the beginning of the COVID-19 pandemic in the UK. To the best of our knowledge, no other research in this area has explicitly examined the persuasiveness of actual public health communications and/or compared them to an alternative. This makes this study unique and of high importance, as it provides crucial insights that can be applied in the case of future crises. A particular aim of the current study was to explore loss versus gain framing. The UK government used a loss frame (more negative focus) in its public health communications and we hypothesised that a gain frame may have been more appropriate, given that a gain frame can motivate cooperation [9,13].

This project investigated the persuasiveness of two public health communications publicised by the UK government (study 1) and compared one of them to an alternative that was adapted to be consistent with a gain frame (study 2). Measuring reactions to actual public health communications has an advantage over using hypothetical (often text-only materials) ones, that have been used by a few other authors (e.g., [32]). Most importantly, such stimuli are realistic and more engaging, as they combine written text with visual elements [34]. In the current study, we examined people’s emotional reactions (PA and NA) and message attitude, as well as willingness to follow the health guidance after seeing a randomly presented COVID-19 public health communication. A selection of theoretically relevant individual differences were also examined, to test their effects on both the persuasiveness of the public health communications as well as compliance with the health guidance. This second part fits well with recent calls suggesting that a more nuanced approach rather than only observing the direct effects of public health communications is needed. For example, “examining more closely the moderators that may augment or attenuate effects [of public health communication], as well as the mediators that can intervene between frame exposure and persuasive effect” ([41] p. 1108) has been suggested as critical to investigate [20]. It is understood that only then can the deep understanding of public health communications be effectively applied.

The current results showed that there was little difference in the perception of public health communications examined. The actual public health communications used in study 1 did not differ on any of the measured variables (i.e., message attitude, PA, NA or behavioural inclination to follow COVID-19 health guidance). Thus, hypothesis H1 was not confirmed in study 1. This is likely due to participants not really perceiving the messages on governmental public health communications as varying to a significant degree. They both mixed positive and negative frames, but it could also be the effect of the “strength of situation”, which is explained more below.

Whereas the loss- vs. gain-framed public health communications from study 2 differed only in the level of negative affect triggered (partial support for H1 here). Specifically, loss-framed governmental public health communication led to a higher level of NA than the gain-framed alternative. Those findings fit with quite a substantial literature suggesting no difference between loss vs. gain frames (e.g., see the meta-analyses in; [41,62]). But they could also be explained by extrapolating past findings ([63]; breast cancer campaign) showing that people with lower susceptibility seem to respond similarly to both frames. Although COVID-19 susceptibility did not show significant effects in our study (apart from significant positive relationship with NA, in study 2), the narrative presented through various media during the pandemic (suggesting very low risk of serious consequences of COVID-19, unless one was elderly or with an underlying health condition) might have contributed to this. It is also likely that the situational context has overridden the framing effects. Indeed “strong situations” [64,65,66] like a pandemic have been shown to exert pressure on people to behave in a certain way, regardless of other factors (e.g., personal characteristics). In other words, “strong situations” restrict variability of behaviour, which we suggest can be seen in our results.

Our results seem to corroborate previous findings showing that adding visual elements to messages overwrites the differences that can be found with text-only communication (e.g., [24]). As can be seen in Figure 1, the public health communications used in the current project incorporated both and were very vivid. However, to be certain that this is the case, the text-only version of the public health communications used here would need to be tested in future studies. Nevertheless, this highlights why it is of paramount importance to study actual, real-world examples of public health communications, as rules guiding reactions to them might be more nuanced than the ones one can observe when using, for instance, text-only and/or hypothetical stimuli.

This study also adds to the literature by providing more evidence (supporting H2) for the role of affect triggered by public health communications. This role has been highlighted as particularly crucial in the literature (e.g., [40]). Recent, systematic evidence demonstrated that loss and gain frames elicit emotions (while gain frames elicit PA, loss frames trigger its counterpart) and that these particular emotions strengthen the persuasive effects of framing [41]. In line with this, in study 2, we found that loss-framed public health communication produced a higher level of NA than gain-framed communication (although there was no difference in PA). The same authors, however, concluded that generally loss and gain frames do not differ in their direct persuasive effect. Due to that, the most impactful effect that gain/loss framing may have could stem from the emotional responses they produce, which subsequently shape one’s behavioural reactions. This is indeed what we found in our studies, specifically that both PA and NA (in study 1; or just PA in study 2) mediated the effect of public health communication attitude onto behavioural inclination to follow COVID-19 guidance. Whereas loss/gain framing did not show any effect (in study 2).

The framing used in study 2 did not show an effect on behavioural inclination to follow COVID-19 guidance. But it is important to highlight that the distribution of data in study 2 revealed few negative opinions regarding intention to follow COVID-19 guidance, as a result our analyses focused on positive inclinations to follow public health communication guidance. The lack of negative opinions most likely reflects the data being collected in the early stages of the COVID-19 pandemic, at which point people’s opinions were unlikely to be influenced by, for example, the length of restrictions, or the perception of the government’s handling of the pandemic. Instead, in the early stages of a crisis situation, people’s willingness to follow health guidance that is aimed at their protection against a threat that is unknown is generally high, and thus public health communications might have limited effect on perceptions to follow advice. Nevertheless, we showed that affect plays a mediational role in one’s willingness to follow COVID-19 guidance, with both PA and NA (although only in study 1, which had bigger outcome variance) partially explaining the link between message attitude and said willingness.

Lastly, we examined the moderating effects of a few individual characteristics of interest and showed that the link between public health communication attitude (study 1) and willingness to follow COVID-19 guidance was moderated by threat minimisation and satisfaction with the government’s handling of the pandemic (partially in line with H3). Specifically, a more positive public health communication attitude led to higher willingness to follow COVID-19 guidance, for all the levels of threat minimisation and government satisfaction, but it was the strongest among those with low threat minimisation and low government satisfaction. While the first result is in line with our hypothesis H3, the latter is somewhat counter-intuitive at first. We think that this latter effect occurred in the direction it did because those with low government satisfaction did not trust the government to deal with the situation well, and thus after seeing the public health communication were strongly motivated to protect themselves and follow health guidance anyway (even though it stemmed from the government that they were not very satisfied with). Additionally, government satisfaction also moderated the link between public health communication attitude and NA triggered. This time, in the predicted direction, with lowest NA among those with high government satisfaction. Interestingly there was no significant effect of COVID-19 susceptibility (partially against H3). A similar analysis testing potential moderation of the relationship between public health communication condition (study 2) and NA demonstrated a significant moderation by threat minimisation only (partially supporting H4). This result showed that for people who minimised the threat of COVID-19, framing did not affect their NA. It is possible then that framing has an effect when there is an optimal level of threat, that is not too high but also not too low (as this can lead to people ignoring/not caring about an issue or distorting their perceptions of threat, which then can result in no framing effects). Overall, these results (partially supporting H3 and H4) suggest that individual characteristics (especially those related to coping with threat—threat minimisation, and satisfaction with authorities responsible for public health communications and tackling crisis) might indeed affect one’s reactions to public health communications and should be at least considered when developing public health communications. However, establishing what factors, in relation to perception of public health communications, affect said perception consistently, or under what conditions, is an avenue for further inquiry. Replications of our findings are indeed needed, especially because the evidence we presented stems from studies with substantial but not large samples.

### 6.1. Limitations

The results of this study are subject to certain limitations. First, the data collected were entirely based on self-report, which, taking into account the normative nature of following health guidance, risks the potential of overreporting willingness to follow COVID-19-mitigating behaviours. This, however, should be somewhat limited by the online, and thus anonymous, nature of the data collection. Second, we only examined willingness to follow the health guidelines and not actual behaviours. Thus, even though the results are applicable to people’s intentions, they do not have to necessarily extend to behavioural level. This said, in many models of behaviour change, intention is seen as a necessary step (e.g., [67,68]) that directly precedes behaviour. Examining intentions is also a highly common practice in literature. Third, it is likely that participants in the current studies came across the two public health communications that were investigated beforehand (both posters in study 1 and one poster in study 2), making this study quasi-experimental. This means that there might be confounding variables that could have affected the reactions that participants had (that we could not control), as some habituation effects could have taken place if participants had seen those adverts before. However, as the study took place when the UK was in the “stay at home” order, frequent exposure to those adverts should have been limited. Fourth, we need to point out that the sample size in our studies was somewhat limited (*n*~150 per study). We strived to recruit a sufficient sample size but had to consider practical constraints such as cost and the urgent timing of the data collection. The sample recruited might have affected the likelihood of finding support for our hypotheses should they rely on a small effect size. However, the analyses we planned to run took this limitation into consideration, and examined a number of variables for which the sample size we had should not have caused an issue. We also presented our results with 95% CI, which provides additional information on the results’ confidence level and is not strongly affected by the sample size. The somewhat limited sample size necessitates utilisation of more substantial samples in future studies. We want to highlight though that similar studies with bigger samples (*n* = 15,929: [69]; *n* = 500: [70]) did not find framing effects either, which corroborates our results and aligns with our argument above that it is highly likely that in “strong” situations (such as the early stages of the COVID-19 pandemic) the situational context matters the most for driving people’s reactions and supersedes other influences (such as framing effects). Lastly, even though we measured whether our participants (or their family members) experienced COVID-19 symptoms, we did not ask about cases of death related to COVID-19. Such an instance would arguably have a profound effect on one’s reaction to COVID-19 public health communications and satisfaction with the UK government’s handling of the pandemic. Future studies might want to consider this omission. However, it is worth noting that a relatively small number of participants in our studies reported experiencing COVID (themselves or within their family), specifically under 25% (study 1) and under 11% (study 2), and thus it is highly likely that a very small number of our participants could have experienced a COVID-19-related death of someone close to them. The estimated percentage of the UK population that died of COVID-19 is around 0.32% (when using 3247.847 death cases per 1 million of UK population, which is around 69.14 million) [71]. This would make reliable analysis of the effects of this factor in our current study not feasible.

### 6.2. Future Directions

It would be interesting to examine whether the results found in the UK are replicated in other countries that used different public health communications or tackled the pandemic in other ways. In relation to the role of satisfaction with the handling of the pandemic, future research could explore broader aspects of this, which could be enabled by using a newly published measure [72]. Additionally, there is a clear need to explore how various individual characteristics affect the persuasiveness of real-life public health communications, especially the emotions triggered and subsequent behavioural effects. More research looking into the effects of specific emotions, such as fear, hope, disgust, guilt, etc., seems to be a promising further avenue. Especially, because they should allow for making more fine-grained predictions. Such a nuanced approach might be particularly important when working on designing public health communications that use techniques other than loss/gain framing techniques. A study [73] showing a simple exchange message being the most effective in increasing hand-washing intentions (via triggered PA) might be of particular interest. In addition, examining the role of source credibility on public health communication’s effectiveness seems to be a topic worth following up on. It would be interesting to see if countries that find their governments more vs. less credible in terms of handling crises react to public health communications differently, and whether a change in government affects this in any way. It is possible that such an effect would be in line with the commonly believed notion that “Trust is hard-earned, easily lost, and difficult to reestablish” [74].

### 6.3. Implications

This study presents some key insights that have implications for using public health communications during health crises. Specifically, it seems that during unprecedented times the way one communicates (i.e., the language used) is not as important as the emotions triggered by said communication. It seems then that a key to promoting citizen compliance is to produce public health communications that have the highest chance of producing PA in the biggest segments of the target audience. Therefore, the characteristics of that audience and techniques leading to PA are worth further investigating. Secondly, although more challenging, the government can improve its chances of people following its guidance by investing time and effort in really making sure that the citizens feel satisfied with its handling of health crises and trust their government (e.g., [75]). This is not an easy task, but it has definite value. One needs to be aware that even though crises can bring a “rally around the flag” effect (i.e., an increase in support for political leaders in the face of crisis) [76], this does not necessarily occur in every country, and if it does, it seems to fade away quickly, which also applied to the COVID-19 pandemic [77] (and was also the case for UK) [78]. According to analyses of the UK government’s handling of the pandemic, PM Boris Johnson was one of the leaders with the lowest citizen support (34% in January 2022; a similar level as the Brazilian Bolsonero who also “downplayed” COVID-19) [79], whereas his government’s public communication and the behaviour of some of its members has been deemed as a case study of “how NOT to communicate in a crisis” [80]. The UK’s PM also experienced the largest drop in approval ratings of any leader of any major economy [81]. Thankfully, it appears that in times of crisis, people can rely on other sources of information (e.g., the BBC) [82], which might limit the negative effects of poorly executed public health communications on citizen compliance and other failures of the government when responding to the pandemic. This finally highlights how crucial a role in communication with the public other non-governmental organisations might have. It is important to examine in the future whether there is a strong effect differentiating the same public health communications should they be released by the government vs. other credible non-governmental organisations.

## Figures and Tables

**Figure 1 ijerph-22-00030-f001:**
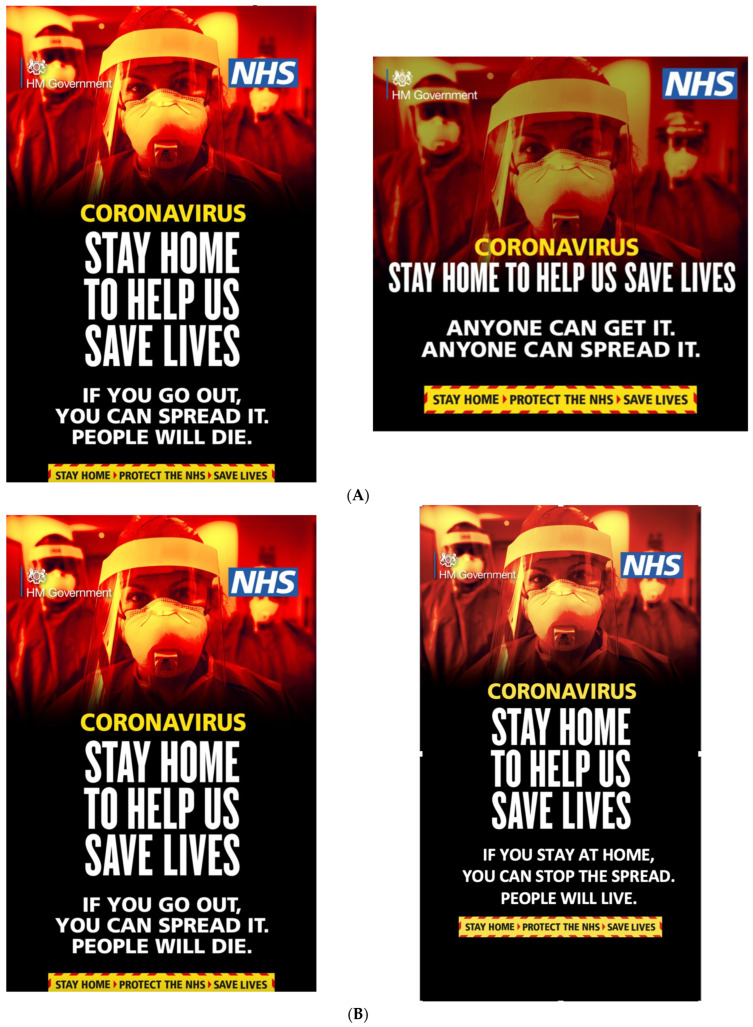
(**A**) The two public health communications used in study 1 (published by the UK government). (**B**) The two public health communications used in study 2.

**Figure 3 ijerph-22-00030-f003:**
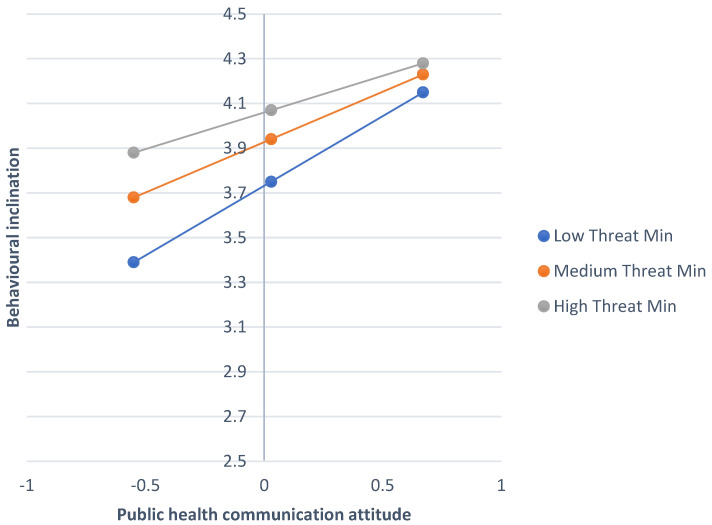
The interaction effect between public health communication attitude and threat minimisation when predicting behavioural inclination to follow guidelines during COVID-19 pandemic.

**Figure 4 ijerph-22-00030-f004:**
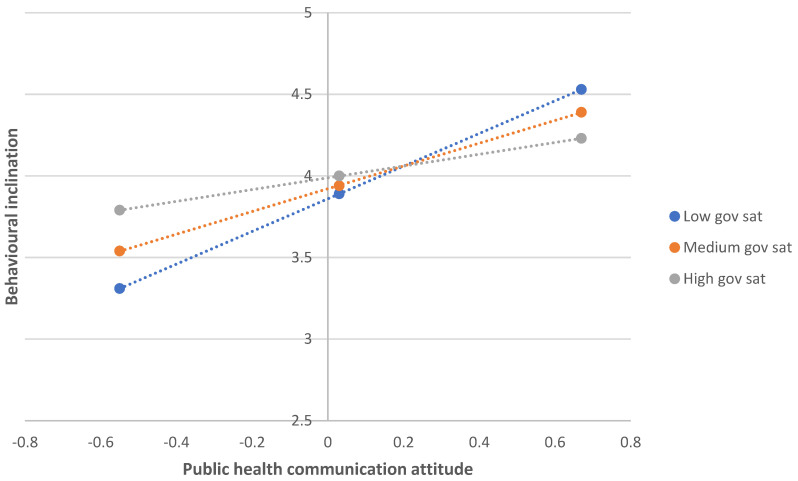
The interaction effect between public health communication attitude and government satisfaction, when predicting behavioural inclination to follow COVID-19 pandemic.

**Figure 5 ijerph-22-00030-f005:**
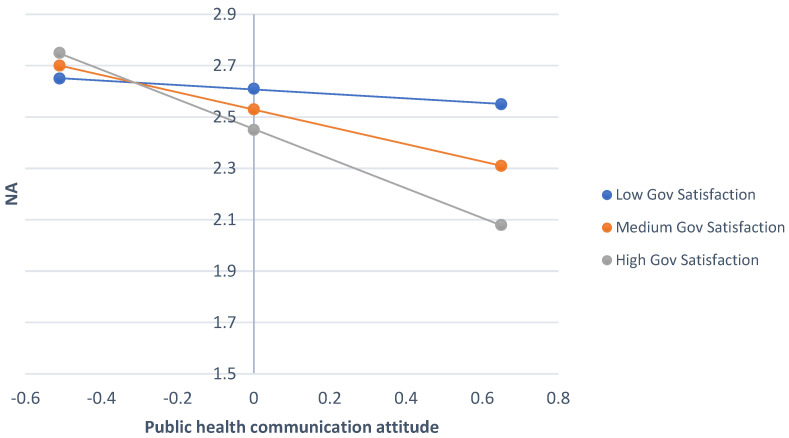
The interaction effect between public health communication attitude and government satisfaction when predicting NA.

**Figure 6 ijerph-22-00030-f006:**
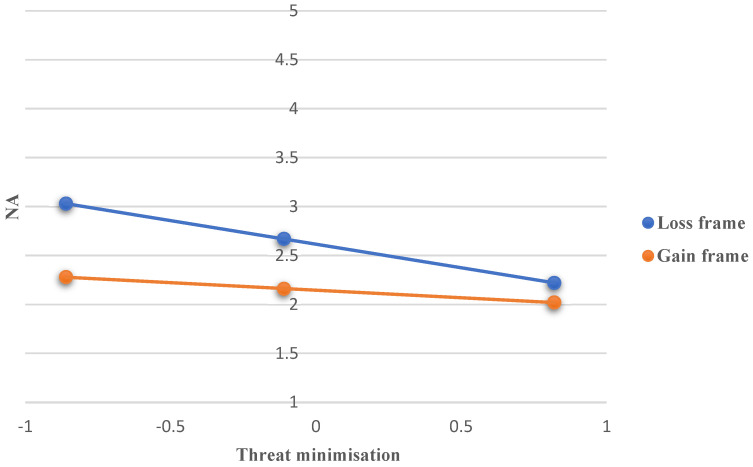
The interaction effect between public health communication condition (loss vs. gain frame) and threat minimisation when predicting NA.

**Table 1 ijerph-22-00030-t001:** Frequencies regarding various variables of interest in study 1.

Variable	*n* (%)	*n* (%)	*n* (%)	*n* (%)	*n* (%)
Following advice on hand washing	No 1 (0.7)	Yes 137 (99.3)			
Following advice on social distancing	No 3 (2.2)	Yes 135 (97.8)			
Following advice on going out only for essential reasons	No 12 (8.7)	Yes 126 (91.3)			
Likelihood of complying with PHC guidance	Not at all likely 6 (4.3)	A little 4 (2.9)	Somewhat likely 14 (10.1)	Mostly likely 68 (49.3)	Extremely likely 45 (32.6)
Change in behavioural inclination after seeing PHC	Decreased a lot 2 (1.4)	Decreased a bit 1 (0.7)	Neither 57 (41.3)	Increased a bit 54 (39.1)	Increased a lot 23 (16.7)
Satisfaction with government COVID-19 handling	Not at all 41 (29.7)	A little 47 (34.1)	Somewhat 29 (21.0)	Very much 17 (12.3)	Extremely 4 (2.9)
COVID-19 symptoms—oneself	Yes 25 (18.1)	No 113 (81.9)			
COVID-19 symptoms—family	Yes 34 (24.6)	No 104 (75.4)			

**Table 2 ijerph-22-00030-t002:** Descriptive statistics for the variables examined in study 1 (*n* = 138).

Variable	Mean	SD	Min	Max	Reliability Index
Regarding public health communication					
Message attitude	3.12	0.64	1.14	5	0.88
PA	2.33	0.84	1	4.2	0.89
NA	2.46	0.81	1	4.7	0.89
Behavioural inclination	3.87	0.77	1.0	5	0.48 ^#^
Individual differences					
Threat minimisation tendency	3.85	0.71	1	5	0.59
Personal risk perception	3.13	0.85	1	5	0.79
Approach motivation	3.31	0.53	2.25	4.83	0.73

Note: ^#^ Pearson’s r coefficient was used as a reliability measure of this 2-item variable. These values are representative of the sample as a whole (i.e., not separated by condition).

**Table 3 ijerph-22-00030-t003:** Comparison of persuasiveness and behavioural inclination of the public health communication conditions used in study 1.

Variable	More Negative Frame (*n* = 66)		Less Negative Frame (*n* = 71)		Test Statistic	*p* Value
	M	SD	M	SD		
Message attitude	3.14	0.63	3.10	0.66	t(134.92) = −0.36	0.72
PA	2.35	0.80	2.31	0.87	t(135) = −0.29	0.77
NA	2.52	0.77	2.40	0.85	t(134) = −0.85	0.40
Behavioural inclination	3.93	0.70	3.80	0.83	t(135) = −0.98	0.33

Note: Behavioural inclination measure is a compound of the two items used in the study (see Methods section for more info).

**Table 4 ijerph-22-00030-t004:** Partial correlations between study 1 variables (controlling for public health communication condition).

Variable	Age	Approach Motiv.	Gov Satisf.	Susc.	Threat Min.	PA	NA	Public Health Communication Attitude	Beh. Incl.
Gender	0.03	−0.01	−0.03	0.01	0.10	−0.19 *	−0.05	−0.07	−0.01
Age		0.12	−0.001	0.12	0.01	0.02	−0.08	−0.06	0.02
Approach motivation			0.12	−0.22 *	−0.20 *	0.15	0.07	−0.07	0.07
Gov satisfaction				−0.21 *	−0.23 **	0.29 **	−0.26 **	0.36 **	0.19 *
Susceptibility					0.37 **	0.02	0.17 #	0.02	0.09
Threat minimisation						0.06	0.24 **	0.16	0.40 **
PA							0.14	0.51 **	0.52 **
NA								−0.34 **	0.15
Public health communication attitude									0.50

Note: # *p* = 0.054, * *p* < 0.05, ** *p* < 0.01.

**Table 5 ijerph-22-00030-t005:** Results of significant moderation analyses for public health communication attitude and individual characteristics, while predicting behavioural inclination to follow COVID-19 guidance and NA.

Outcome	Level	Effect	SE	t	*p*	95% CI
Behavioural inclination to follow COVID-19 guidance	Threat minimisation (R^2^ = 0.43)					
	Low	0.63	0.08	7.50	<0.001	0.46; 0.79
	Medium	0.45	0.09	4.88	<0.001	0.27; 0.63
	High	0.33	0.12	2.81	<0.006	0.10; 0.56
	Satisfaction with government’s handling of COVID-19 pandemic (R^2^ = 0.37)					
	Low	1.00	0.13	7.97	<0.001	0.75; 1.25
	Medium	0.69	0.09	7.70	<0.001	0.52; 0.87
	High	0.36	0.11	3.26	0.001	0.14; 0.58
NA	Satisfaction with government’s handling of COVID-19 pandemic (R^2^ = 0.19)					
	Low	−0.09	0.16	−0.56	0.57	−0.41; 0.23
	Medium	−0.33	0.11	−2.92	0.004	−0.56; −0.11
	High	−0.58	0.14	−4.26	<0.001	−0.85; −0.31

**Table 6 ijerph-22-00030-t006:** Frequencies regarding various variables of interest in study 2.

Variable	*n* (%)	*n* (%)	*n* (%)	*n* (%)	*n* (%)
Political orientation	Very liberal 13 (8.2)	Liberal 63 (39.9)	Middle of the road 61 (38.6)	Conservative 21 (13.3)	Very conservative 0 (0)
Following advice on hand washing	Not at all 1 (0.6)	A little 5 (3.2)	Somewhat 14 (8.9)	A lot 53 (33.5)	Extremely 85 (53.8)
Following advice on social distancing	Not at all 0 (0)	A little 7 (4.4)	Somewhat 6 (3.8)	A lot 55 (34.8)	Extremely 90 (57.0)
Following advice on going out only essential reasons	Not at all 2 (1.3)	A little 7 (4.4)	Somewhat 21 (13.3)	A lot 49 (31.0)	Extremely 79 (50.0)
Supporting Conservative party in last elections	Yes 30 (19.0)	No 114 (72.2)	Prefer not to say 14 (8.9)		
Satisfaction with government’s COVID-19 handling	Not at all 41 (25.9)	A little 52 (32.9)	Somewhat 48 (30.4)	A lot 14 (8.9)	Extremely 3 (1.9)
COVID-19 symptoms—oneself	Definitely yes 4 (2.5)	Probably yes 21 (13.3)	Rather not 17 (10.8)	Definitely not 116 (73.4)	
COVID-19 symptoms—family	Definitely yes 18 (11.4)	Probably yes 18 (11.4)	Rather not 13 (8.2)	Definitely not 109 (69.0)	

**Table 7 ijerph-22-00030-t007:** Descriptive statistics for the variables examined in study 2 (*n* = 158).

Variable	Mean	SD	Min	Max	Reliability Index
Regarding public health communication					
Message attitude	3.28	0.63	1.43	5	0.87
PA	2.41	0.80	1	4.6	0.87
NA	2.39	0.84	1	4.5	0.89
Behavioural inclination	4.49	0.65	1.6	5	0.90
Individual differences					
Threat minimisation tendency	2.36	0.84	1	5	0.70
Personal risk perception	2.98	0.82	1	5	0.71
Trait reactance	2.51	0.67	1	4.73	0.83
Trust in government	2.35	0.96	1	5	0.83
Citizen compliance	3.26	0.77	1	5	0.66

**Table 8 ijerph-22-00030-t008:** Comparison of persuasiveness and behavioural inclination of the public health communications used in study 2.

	Loss Frame (*n* = 80)		Gain Frame (*n* = 78)			
Variable	M	SD	M	SD	Test Statistic	*p*
Message attitude	3.23	0.65	3.33	0.62	t(156) = 0.62	0.54
PA	2.32	0.75	2.50	0.85	t(155.91) = −1.03	0.30
NA	2.66	0.86	2.13	0.75	t(156) = 4.09	<0.001
Behavioural inclination	83.79 *	-	75.10 *	-	U = 2776.50	0.21

* Mann-Whitney test used due to the distribution of the data.

**Table 9 ijerph-22-00030-t009:** Partial correlations between study 2 variables (controlling for public health communication condition).

Variable	Age	Trait React.	Pol. Ori.	Gov Trust	Gov Compl.	Susc.	Threat Min.	PA	NA	Public Health Communication Att.	Beh. Incl.
Gender	0.10	−0.09	0.13	0.06	0.09	0.15	−0.06	−0.01	0.11	−0.15 #	−0.09
Age		0.04	0.12	0.001	−0.04	0.04	0.06	−0.01	−0.15	−0.03	0.14
Trait reactance			0.05	−0.07	−0.32 **	−0.15 #	0.23 **	−0.13	0.13	−0.18 *	−0.10
Political orientation				0.34 **	0.28 **	0.01	0.14	0.10	0.17 *	−0.07	0.05
Gov trust					0.51 **	−0.05	0.06	0.15 #	0.04	0.05	0.11
Gov compliance						0.05	−0.07	0.17 *	0.01	0.19 *	0.22 **
Susceptibility							−0.46 **	0.16 #	0.36 **	−0.11	0.11
Threat min.								−0.18 *	−0.33 **	−0.05	−0.12
PA									0.15 #	0.49 **	0.36 **
NA										−0.35 **	−0.01
Public health communication attitude											0.28 **

Note: ** *p* < 0.001, * *p* < 0.05, # *p* < 0.07; behavioural inclination and gender were dummy-coded as “0”, “1”.

**Table 10 ijerph-22-00030-t010:** The results of binary logistic regression predicting behavioural inclination to follow COVID-19 guidance.

Variable	B	SE	Wald Statistic	*p* Value	Exp(B)	95% CI
Step 1	X^2^ = 6.84, *p* = 0.08	Nagelkarke R^2^ = 0.06	56.4% classified			
Constant	−1.67	0.61	7.64	0.006	0.19	-
Public health communication condition	0.29	0.34	0.71	0.40	1.33	0.69–2.57
Age	0.04	0.02	4.84	0.028	1.04	1.00–1.07
Sex	0.44	0.37	1.39	0.24	1.55	0.75–3.22
Step 2	X^2^ = 11.79 *p* = 0.001	Nagelkarke R^2^ = 0.16	64.4% classified			
Constant	−5.22	1.31	16.01	<0.001	0.01	
Public health communication condition	0.37	0.35	1.09	0.30	1.45	0.72–2.89
Age	0.04	0.02	5.58	0.02	1.04	1.00–1.08
Sex	0.30	0.39	0.58	0.45	1.34	0.63–2.88
Message attitude	1.02	0.32	10.34	0.001	2.77	1.49–5.16
Step 3	X^2^ = 12.16, *p* = 0.002	Nagelkarke R^2^ = 0.25	69.8% classified			
Constant	−7.20	1.86	15.03	<0.001		
Public health communication condition	0.43	0.41	1.11	0.29	1.54	0.69–3.41
Age	0.05	0.02	6.09	0.01	1.05	1.01–1.09
Sex	0.47	0.42	1.30	0.25	1.61	0.71–3.62
Message attitude	0.75	0.41	3.38	0.07	2.12	0.95–4.70
PA	0.82	0.31	7.23	0.007	2.28	1.25–4.16
NA	0.25	0.28	0.81	0.37	1.29	0.75–2.22

**Table 11 ijerph-22-00030-t011:** Results of moderation analyses for public health communication condition and individual characteristics, while predicting NA.

Key Variables	Coeff.	SE	t	*p*	95% CI
Trait reactance (R^2^ = 0.15, *p* < 0.001)					
X	−0.53	0.13	−4.22	<0.001	−0.78; −0.28
M	0.17	0.13	1.30	0.19	−0.09; 0.43
X × M	0.01	0.19	0.04	0.97	−0.38; 0.39
COVID-19 susceptibility (R^2^ = 0.25, *p* < 0.001)					
X	−0.47	0.12	−3.92	<0.001	−0.71; −0.23
M	0.36	0.10	3.46	<0.001	0.15; 0.56
X × M	−0.01	0.15	−0.05	0.96	−0.30; 0.28
Threat minimisation (R^2^ = 0.25, *p* < 0.001) **					
X	−0.47	0.12	−3.90	<0.001	−0.70; −0.23
M	−0.49	0.11	−4.47	<0.001	−0.70; −0.27
X × M	0.33	0.14	2.31	0.02	0.05; 0.62
Trust in the government (R^2^ = 0.14, *p* < 0.001)					
X	−0.53	0.13	−4.14	<0.001	−0.78; −0.28
M	−0.02	0.10	−0.15	0.88	−0.21; 0.18
X × M	0.07	0.14	0.54	0.59	−0.19; 0.34
Political orientation (R^2^ = 0.16, *p* < 0.001)					
X	−0.56	0.13	−4.41	<0.001	−0.81; −0.31
M	0.16	0.11	1.41	0.16	−0.06; 0.38
X × M	0.02	0.15	0.12	0.90	−0.28; 0.32
Past citizen compliance (R^2^ = 0.13, *p* < 0.001)					
X	−0.53	0.13	−4.12	<0.001	−0.78; −0.27
M	<0.01	0.12	<0.01	1.00	−0.24; 0.24
X × M	−0.01	0.17	−0.05	0.96	−0.34; 0.32

Note: X = public health communication condition (loss/gain frame), M = moderator (as specified), X × M = interaction term; ** denotes significant moderation.

## Data Availability

The data are available upon request from the authors.

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
