# Peer review of "Persuasiveness of Public Health Communication During the COVID-19 Pandemic: Message Framing, Threat Appraisal, and Source Credibility Effects"

_ijerph, 2024, doi:10.3390/ijerph22010030_

Round 1
Reviewer 1 Report
Comments and Suggestions for Authors
This is quasi-experimental rather than experimental research.
Study 1, coinciding with the period of display of posters of the campaign, would be conditioned by the possible recall of the campaign in progress, while the possible effects of the messages of study 2, since they are prepared ad hoc, as an alternative to the actual messages broadcast, would not have the conditioning of the memory of the message, presented as novel. An issue presented in 6.1. of Limitations.
It would be more convenient to formulate the hypotheses in present or future, instead of using the conditional mode (H2, H3, H4).
You ask about demographic data and your current protective behavior against COVID-19, as well as the presence of symptoms on yourself or your family members, but there is a key question that would determine the susceptibility in the perception of risk and fear of contagion, such as whether it is known of any case of death of a close person by Covid-19.
It is not the same perception of susceptibility to contracting the Covid, as the certainty of the nefarious effect of death in a close acquaintance.
¡Excellent paper!
Author Response
|
Response to Reviewer 1 Comments
|
||
|
1. Summary |
|
|
|
Thank you very much for taking the time to review this manuscript. Please find the detailed responses below and the corresponding revisions/corrections highlighted in the re-submitted files.
|
||
|
2. Questions for General Evaluation |
Reviewer’s Evaluation |
Response and Revisions |
|
Does the introduction provide sufficient background and include all relevant references? |
Yes |
Thank you for grading our manuscript highly. |
|
|
|
|
|
Is the research design appropriate? |
Yes |
|
|
Are the methods adequately described? |
Yes |
|
|
Are the results clearly presented? |
Yes |
|
|
Are the conclusions supported by the results? |
Yes |
|
|
3. Point-by-point response to Comments and Suggestions for Authors |
||
|
Comment 1: This is quasi-experimental rather than experimental research. Study 1, coinciding with the period of display of posters of the campaign, would be conditioned by the possible recall of the campaign in progress, while the possible effects of the messages of study 2, since they are prepared ad hoc, as an alternative to the actual messages broadcast, would not have the conditioning of the memory of the message, presented as novel. An issue presented in 6.1. of Limitations. |
||
|
Response 1: Thank you for pointing this out. We agree with this comment. There is a possibility that some participants in study 1 could have been exposed to the posters before (albeit in both groups), and that in study 2 participants in real-life but not in the adapted poster condition could have seen it before. Therefore, we have highlighted that this is a quasi-experimental study in the relevant sections of the manuscript. We have also highlighted this limitation in the discussion.
[page 25, paragraph 6.1, and line 836-843] “[Third, it is likely that participants in the current studies came across the two PHCs that were investigated beforehand (both posters in study 1 and one poster in study 2), making this study quasi-experimental. This means that there might be confounding variables that could have affected the reactions that participants had (that we could not control), as some habituation effects could have taken place if participants had seen those adverts before. However, as the study took place when the UK was in the “stay at home” order frequent exposition to those ads should be limited.]” |
||
|
|
||
|
Comment 2: It would be more convenient to formulate the hypotheses in present or future, instead of using the conditional mode (H2, H3, H4). Response 2: Agree. We have modified the tense of hypotheses H2-H4 accordingly.
[page 5, paragraph 2.3, and line 201-219] “[H2: Both PA (positively) and NA (negatively) will mediate the effect of the PHC attitude on one’s behavioural inclination to follow Covid-19 guidelines. H3: More/less negative PHC’s effect on PHCs reactions (NA, PA, and PHC attitude) will be moderated by individual characteristics, with more negative PHC showing more negative effect and strengthening with high Covid-19 susceptibility, low gov satisfaction, and low threat minimisation. H4: The loss vs. gain-frame effect on PHCs reactions (NA, PA, and PHC attitude) will be moderated by individual characteristics, with loss frame showing more negative effect, and strengthening with low threat minimisation, high trait reactance, high Covid-19 susceptibility, and low level of governmental support (i.e., low satisfaction with government’s handling of the pandemic, low past compliance with gov policies, and political orientation opposite to the current government)]”
Comment 3: You ask about demographic data and your current protective behaviour against COVID-19, as well as the presence of symptoms on yourself or your family members, but there is a key question that would determine the susceptibility in the perception of risk and fear of contagion, such as whether it is known of any case of death of a close person by Covid-19. It is not the same perception of susceptibility to contracting the Covid, as the certainty of the nefarious effect of death in a close acquaintance. Response 3: Agree. We added a comment about this to the discussion to emphasize this point. We will try to look into this in our follow-up studies. It is worth noting though that a relatively small number of participants in our studies reported experiencing Covid (themselves or within their family), specifically under 25% in study 1, under 11% in study 2, and thus it is highly likely that a very minute number of our participants could have experienced Covid-19 related death of someone close to them. The national level of Covid-19-related deaths also highlights that it is unlikely that a substantial number of our participants would have experienced such an event. We explained this in the text in the discussion (see below).
[page 26, paragraph 6.1, and line 857-870] “[Lastly, even though we measured whether our participants (or their family members) experienced Covid-19 symptoms, we did not ask about cases of death related to Covid-19. Such an instance would arguably have a profound effect on one’s reaction to Covid-19 PHCs and the satisfaction with the UK government handling of pandemic. Future studies might want to consider this omission. However, it is worth noting that a relatively small number of participants in our studies reported experiencing Covid (themselves or within their family), specifically under 25% (study 1) and under 11% (study 2), and thus it is highly likely that a very minute number of our participants could have experienced Covid-19 related death of someone close to them. The estimated percentage of UK population that died of Covid-19 is around 0.32% (when using 3247.847 death cases per 1 million of UK population which is around 69.14 million (Our World in Data, 2023). This would make reliable analysis of the effects of this factor in our current study not feasible.
Comment 4: ¡Excellent paper!
Response 4: Thank you for this very encouraging feedback. We really appreciate it.
|
||
|
4. Response to Comments on the Quality of English Language |
||
|
Point 1: The quality of English does not limit my understanding of the research |
||
|
Response 1: Thank you for this comment, we appreciate that the quality of English language in our manuscript doesn’t limit its understanding. |
||
|
5. Additional clarifications |
||
|
NA. We want to express our thanks to the reviewers again and hope that the way we addressed their comments is satisfactory. We deeply believe that those changes have made our manuscript of better quality. We also really appreciate the very positive feedback that was made about the paper being “excellent” and providing an “intriguing” discussion.
Sincerely, Dr Natalia Stanulewicz-Buckley (first author) |
||
Reviewer 2 Report
Comments and Suggestions for Authors
Review report on "Persuasiveness of public health communication during the COVID-19 pandemic: message framing, threat appraisal, and source credibility effects."
This paper studies the relative effectiveness of the UK governments public health messages used during the first wave of the COVID-19 pandemic. The paper compares messages using a loss and a gain frame.
The paper has an essential contribution to the debate on public health messages and uses an exciting method to perform an experiment and test what's more effective. Below, I point out a few comments and suggestions to improve the paper:
- The paper uses a "representative" sample of UK adults (300). Explain in detail how representative the sample is.
- In the paper, we have the sentence, "The study was approved by the [blank] ethics committee." The paper needs to fill in the blanks, as readers need to know which ethics committee approved the research.
- Explain in more detail the parallel multiple mediator model. Explain the regression and show the equation with all variables included in the specification.
- Put an appendix or supplementary material in the paper with both studies to comprehensively analyze all the information gathered.
- The paper discusses the limitations of the research, with which I agree. However, are there any limitations regarding the statistical analysis as you have approximately 150 respondents in each study? I wonder if you had a larger sample results would be stronger. What can be said about that? Provide some discussion on the potential limitations on this front.
- I find the discussion in the last paragraph genuinely fascinating. Suppose the public support for the Boris Johnson administration was low, and his cabinet was a study case on how NOT to communicate in a crisis. In that case, the public may doubt PHC and look for alternatives. The same goes for Bolsonaro (Brazilian president during COVID-19). How surprising are the results? What could be done differently to get more insights in further research?
Author Response
|
Response to Reviewer 2 Comments
|
||
|
1. Summary |
|
|
|
Thank you very much for taking the time to review this manuscript. Please find the detailed responses below and the corresponding revisions/corrections highlighted in the re-submitted files.
|
||
|
2. Questions for General Evaluation |
Reviewer’s Evaluation |
Response and Revisions |
|
Does the introduction provide sufficient background and include all relevant references? |
Can be improved |
Thank you for grading our manuscript highly. We understand your view that certain aspects of it could be improved. We believe that by addressing the comments of our reviewers we have improved those aspects. |
|
Is the research design appropriate? |
Can be improved |
|
|
Are the methods adequately described? |
Can be improved |
|
|
Are the results clearly presented? |
Can be improved |
|
|
Are the conclusions supported by the results? |
(no answer) |
|
|
3. Point-by-point response to Comments and Suggestions for Authors |
||
|
Comment 1: Review report on "Persuasiveness of public health communication during the COVID-19 pandemic: message framing, threat appraisal, and source credibility effects." This paper studies the relative effectiveness of the UK governments public health messages used during the first wave of the COVID-19 pandemic. The paper compares messages using a loss and a gain frame. The paper has an essential contribution to the debate on public health messages and uses an exciting method to perform an experiment and test what's more effective. Below, I point out a few comments and suggestions to improve the paper: 1. The paper uses a "representative" sample of UK adults (300). Explain in detail how representative the sample is.
|
||
|
Response 1: Thank you for pointing this out. We have corrected this information in our paper and changed it to a voluntary sample.
[page 5, paragraph 3.1, and line 225-227] “[This was a voluntary sample of UK residents (ibidem for study 2), recruited by Prolific.”] |
||
|
Comment 2: In the paper, we have the sentence, "The study was approved by the [blank] ethics committee." The paper needs to fill in the blanks, as readers need to know which ethics committee approved the research. |
||
|
Response 2: We are sorry for this omission. We have now completed these sentences accordingly.
[page number 5; 8, paragraph 3.1; 3.2, and line 228; 327] “["The study was approved by the De Montfort University research ethics committee."]”
Comment 3: Explain in more detail the parallel multiple mediator model. Explain the regression and show the equation with all variables included in the specification. Response 3: Thank you for pointing this out. We have, accordingly, modified the parallel multiple mediator model description to make it more detailed. We believe that its current description together with the figure that now includes more information depicts this analysis clearly and is in line with the recommended ways of reporting mediation models (e.g., Best (but oft-forgotten) practices: mediation analysis - PMC). We also added the mediation equations with all the variables.
[page 13, paragraph 4.4.]
Comment 4: Put an appendix or supplementary material in the paper with both studies to comprehensively analyze all the information gathered.
Response 4: We apologise, but we are not certain what material is the reviewer suggesting here. If this can be clarified, we are happy to prepare such an appendix or supplementary material. We want to point out that the PHC posters are already included in the paper, and we provided a detailed description of our methods and results as well.
Comment 5: The paper discusses the limitations of the research, with which I agree. However, are there any limitations regarding the statistical analysis as you have approximately 150 respondents in each study? I wonder if you had a larger sample results would be stronger. What can be said about that? Provide some discussion on the potential limitations on this front. Response 5: We are glad to hear that the reviewer agrees with our discussion of the study’s limitations. We have, however, expanded limitations section to include the suggested discussion of the sample size/study’s power. We have added a comment about the potential of obtaining stronger results, should the sample be bigger, but also highlighted the practical constraints that made it not possible for us to recruit bigger samples. We also added a reference to a similar study with much bigger sample that also didn’t find the loss/gain framing effect.
[page 25-26, paragraph 6.1, and line 843-857] [“Fourth, we need to point out that the sample size in our studies was somewhat limited (n~150 per study). We have strived to recruit a sufficient sample but had to consider practical constraints such as cost and urgent timing of the data collection. The sample recruited might have affected the likelihood of finding support for our hypotheses shall they rely on a small effect size. However, the analyses we planned to run took this limitation into consideration, and examined a number of variables that should not cause an issue due to the sample size we had. We also presented our results with 95% CI, which provides additional information on results’ confidence level that is not affected by the sample size. This somewhat limited sample size remits utilisation of more substantial samples in future studies though. We want to highlight though that similar studies with bigger samples (n=15929: Dorison et al., 2022; n=500: Sanders et al., 2021) did not find framing effects either, which corroborate our results and aligns with our argument above that it is highly likely that in “strong” situations (such as early stages of the Covid-19 pandemic) the situational context matters the most for driving people’s reactions and supersedes other influences (such as framing effects).”]
Comment 6: I find the discussion in the last paragraph genuinely fascinating. Suppose the public support for the Boris Johnson administration was low, and his cabinet was a study case on how NOT to communicate in a crisis. In that case, the public may doubt PHC and look for alternatives. The same goes for Bolsonaro (Brazilian president during COVID-19). How surprising are the results? What could be done differently to get more insights in further research? Response 6: We are really pleased to hear that reviewer 2 has found our discussion fascinating. We do think that Boris Johnson’s handling of the pandemic (with all the errors that came to the public eye in later stages) could be seen as a study case on how NOT to communicate in crisis. We have added a comment to our discussion (future directions section), in line with reviewer 2’s questions above.
[page 26, paragraph 6.2, and line 886-892] “[In addition, examining the role of source credibility on PHCs effectiveness seems to be a topic worth following up on. It would be interesting to see if countries that find their governments more vs. less credible in terms of handling crises react to PHCs differently, and whether a change of government affects this in any way. It is possible that such an effect would go in line with the commonly believed idea that “Trust is hard-earned, easily lost, and difficult to reestablish” (Harmeling, 2021).]
[page 27, paragraph 6.3, and line 919-922] We also added this at the end of implications: [“It would be interesting to examine in future if there would be a strong effect differentiating the same PHCs if they were supposed to be released by the government vs. other credible non-governmental organizations.”]
|
||
|
4. Response to Comments on the Quality of English Language |
||
|
Point 1: The quality of English does not limit my understanding of the research |
||
|
Response 1: Thank you for this comment, we appreciate that the quality of English language in our manuscript doesn’t limit its understanding.
|
||
|
5. Additional clarifications |
||
|
We would like to ask for clarification regarding comment 5 made by the reviewer 2 as we are not sure at the moment what we are being asked to do in that comment.
We want to express our thanks to the reviewers again and hope that the way we addressed their comments is satisfactory. We deeply believe that those changes have made our manuscript of better quality. We also really appreciate the very positive feedback that was made about the paper being “excellent” and providing an “intriguing” discussion.
Sincerely, Dr Natalia Stanulewicz-Buckley (first author) |
||
Reviewer 3 Report
Comments and Suggestions for Authors
This is a very interesting study. Thank you for the opportunity to review it.
I would like to point out the following points that I noticed in order.
1. There seems to be a lot of abbreviations in this paper. In particular, PHC is used frequently, but many readers will mistake it for primary health care. I recommend that you spell out everything as public health communication. The same goes for NA and PA.
2. Observational studies should follow the STROBE statement. STROBE consists of four chapters: Introduction, Method, Results and Discussion. In this paper, I think chapters 1 and 2 can be combined into the Introduction.
3. Is the research design a cross-sectional design or a follow-up design? I think it is probably a cross-sectional design, but please specify.
4. There are various analysis methods. Please clearly explain in the methods chapter that you have selected them in line with the purpose of the study. In the study with behavioral inclination as the dependent variable, linear regression was used in Study 1, but binary logistic regression was used in Study 2. Please give a reason for this.
5. Please clearly explain the basis for your sample size.
6. In your discussion, please state whether Hypotheses 1 to 4 were verified or not, and if not, clearly explain why.
Author Response
|
Response to Reviewer 3 Comments
|
||
|
1. Summary |
|
|
|
Thank you very much for taking the time to review this manuscript. Please find the detailed responses below and the corresponding revisions/corrections highlighted in the re-submitted files.
|
||
|
2. Questions for General Evaluation |
Reviewer’s Evaluation |
Response and Revisions |
|
Does the introduction provide sufficient background and include all relevant references? |
Yes |
Thank you for grading our manuscript highly. We understand your view that the description of the methods must be improved. We made every effort to address this in the updated manuscript. |
|
Is the research design appropriate? |
Yes |
|
|
Are the methods adequately described? |
Must be improved |
|
|
Are the results clearly presented? |
Can be improved |
|
|
Are the conclusions supported by the results? |
Can be improved |
|
|
3. Point-by-point response to Comments and Suggestions for Authors
|
||
|
Comment 1: This is a very interesting study. Thank you for the opportunity to review it. I would like to point out the following points that I noticed in order. There seems to be a lot of abbreviations in this paper. In particular, PHC is used frequently, but many readers will mistake it for primary health care. I recommend that you spell out everything as public health communication. The same goes for NA and PA. |
||
|
Response 1: Thank you for your kind comment about our study being interesting. Thank you for pointing out the potential issue with PHC acronym. We agree with this comment. Therefore, we have spelled out this acronym throughout our manuscript. NA and PA are much more established as acronyms in the emotions literature and so we do not believe they need spelling out.
|
||
|
Comment 2: Observational studies should follow the STROBE statement. STROBE consists of four chapters: Introduction, Method, Results and Discussion. In this paper, I think chapters 1 and 2 can be combined into the Introduction.
|
||
|
Response 2: In line with the comment made by reviewer 1 this study was a quasi-experimental, not an observational study. We do not think that combining chapters 1 and 2 would make a useful change to our paper, as they are both already a part of the introduction.
Comment 3: Is the research design a cross-sectional design or a follow-up design? I think it is probably a cross-sectional design, but please specify.
Response 3: As stated above, this study used a quasi-experimental (between group) design, with an attempt to compare two separate PHCs in study 1 (which ended up being collapsed due to no significant differences found) and study 2 (which did compare two groups/two PHCs). We have highlighted this more clearly in the manuscript now.
[page 2, paragraph 1, and line 80-81.] “[In Section 3 we describe our quasi-experimental methods.]”
[page 5, paragraph 3.1, and line 239-241.] “[Design: This study used a between-subjects quasi-experimental design, where participants were randomly introduced to one of two PUBLIC HEALTH COMMUNICATIONs, see Figure 1A.]”
[page 25, paragraph 6.1, and line 836-843.] [“Third, it is likely that participants in the current studies came across the two PUBLIC HEALTH COMMUNICATIONs that were investigated beforehand (both posters in study 1 and one poster in study 2), making this study quasi-experimental. This means that there might be confounding variables that could have affected the reactions that participants had (that we could not control), as some habituation effects could take place if seeing those adverts before. However, as the study took place when UK was in the “stay at home” order frequent exposition to those ads should be limited.”]
Comment 4: There are various analysis methods. Please clearly explain in the methods chapter that you have selected them in line with the purpose of the study. In the study with behavioral inclination as the dependent variable, linear regression was used in Study 1, but binary logistic regression was used in Study 2. Please give a reason for this.
Response 4: Thank you for pointing this out. We have now stated more clearly that our analysis methods were selected in line with the purpose of the study.
[page 9, paragraph 4, line 385-388] “[All the analyses have been selected in line with the purpose of this study, specifically, comparing responses to PUBLIC HEALTH COMMUNICATIONs, and examining the moderating effect of individual characteristics, as well as mediating effect of affect.]”
[page 16, paragraph 5, line 583-586] [“Following from and expanding on study 1, all the analyses have been selected in line with the purpose of this study, specifically, comparing responses to PUBLIC HEALTH COMMUNICATIONs, and examining the moderating effect of individual characteristics, as well as mediating effect of PUBLIC HEALTH COMMUNICATION attitude and affect.”]
We have also made it more clear why linear regression was used in Study 1, whereas binary regression in Study 2. This was due to the nature of the outcome data – their distribution in study 2 did not allow for running linear regression (highly skewed) and thus binary regression with re-coded data was applied.
[page 19, paragraph 5.3, and line 618-625] “[It is worth highlighting that even though behavioural inclination to follow Covid-19 guidance in Study 1 was analysed with linear regression. In study 2, due to the non-normal distribution and highly skewed data of this main outcome (i.e., behavioural inclination to follow Covid-19 guidelines, which demonstrated negative binomial distribution with overdispersion of highest possible value) for the subsequent analyses this variable was recoded into a binary variable, with those who scored the maximum possible value (5) coded as one group (dummy coded as 1), whereas those who scored above the midpoint (from 3.20 to 4.80) but below maximum as the second group (dummy coded as 0).]”
Comment 5: Please clearly explain the basis for your sample size.
Response 5: We have added an explanation for our sample size as well as highlighted it as a limitation.
[page 5, paragraph 3.1, and line 232-238] [“The sample size was based on the consideration of analyses planned, but also practical considerations (available funds and time urgency for data collection in early pandemic). As the main interest of this study was analysis of differences between more negative/loss and more positive/gain messages, we ran a power analysis to estimate the recommended sample size. The analysis with the power=.80, alpha=.05, and small-to-medium effect size (Cohen’s d=.40), suggested a sample of 156 participants, which we aimed to recruit for both studies.”]
[page 25-26, paragraph 6.1, and line 843-857] [“Fourth, we need to point out that the sample size in our studies was somewhat limited (n~150 per study). We have strived to recruit a sufficient sample but had to consider practical constraints such as cost and urgent timing of the data collection. The sample recruited might have affected the likelihood of finding support for our hypotheses shall they rely on a small effect size. However, the analyses we planned to run took this limitation into consideration, and examined a number of variables that should not cause an issue due to the sample size we had. We also presented our results with 95% CI, which provides additional information on results’ confidence level that is not affected by the sample size. This somewhat limited sample size remits utilisation of more substantial samples in future studies though. We want to highlight though that similar studies with bigger samples (n=15929: Dorison et al., 2022; n=500: Sanders et al., 2021) did not find framing effects either, which corroborate our results and aligns with our argument above that it is highly likely that in “strong” situations (such as early stages of the Covid-19 pandemic) the situational context matters the most for driving people’s reactions and supersedes other influences (such as framing effects).”]
[page 24, paragraph 6, and line 821-825] [“However, establishing what factors in relation to perception of PUBLIC HEALTH COMMUNICATIONS do affect said perception consistently, or under what conditions is an avenue for further inquiry. Replications of our findings are indeed needed, especially because the evidence we presented stems from studies with substantial but not large samples.”]
Comment 6: In your discussion, please state whether Hypotheses 1 to 4 were verified or not, and if not, clearly explain why.
Response 6: Thank you for pointing this out. We have now clearly specified whether our hypotheses were supported by the data or not. We also highlighted more what might be the reasons for not finding support for some of our hypotheses.
[page 23, paragraph 6, and line 734-738] “[Thus, hypothesis H1 was not confirmed in study 1. This is likely due to participants not really perceiving the messages on governmental PUBLIC HEALTH COMMUNICATIONs as varying to a significant degree. They both mixed positive and negative frames. Whereas the loss- vs. gain-framed PUBLIC HEALTH COMMUNICATIONs from study 2 differed only in the level of negative affect triggered (partial support for H1 here).]”
[page 23-24, paragraph 6, and line 748-753] [“It is also likely that the situational context has overridden the framing effects. Indeed “strong situations” (Mischel, 1977; Hough & Oswald, 2008; Li et al., 2024) like pandemic have been shown to exert pressure on people to behave in certain way, regardless of other factors (e.g., personal characteristics). In other words, “strong situations” restrict variability of behaviour, which we propose can be seen in our results.”]
[page 24, paragraph 6, and line 764-778] [“This study also adds to the literature by providing more evidence (supporting H2) for the role of affect triggered by PUBLIC HEALTH COMMUNICATIONs. (…) This is indeed what we found in our studies, specifically that both PA and NA mediated the effect of PUBLIC HEALTH COMMUNICATION attitude onto behavioural inclination to follow Covid-19 guidance (in both study 1 and 2), whereas loss-gain framing did not show any effect (in study 2).]
[page 24, paragraph 6, and line 793-796] [“Lastly, we examined moderating effects of few individual characteristics of interest and showed that the link between PUBLIC HEALTH COMMUNICATION attitude (study 1) and willingness to follow Covid-19 guidance was moderated by threat minimisation and satisfaction with government’s handling of the pandemic (partially in line with H3).”]
|
||
|
4. Response to Comments on the Quality of English Language |
||
|
Point 1: The quality of English does not limit my understanding of the research |
||
|
Response 1: Thank you for this comment, we appreciate that the quality of English language in our manuscript doesn’t limit its understanding.
|
||
|
5. Additional clarifications |
||
|
NA. We want to express our thanks to the reviewers again and hope that the way we addressed their comments is satisfactory. We deeply believe that those changes have made our manuscript of better quality. We also really appreciate the very positive feedback that was made about the paper being “excellent” and providing an “intriguing” discussion.
Sincerely, Dr Natalia Stanulewicz-Buckley (first author) |
||
Round 2
Reviewer 2 Report
Comments and Suggestions for Authors
The paper has been revised, and it can be published.